# A National Survey of Children, Adults, and the Elderly in the Fourth Wave of the COVID-19 Pandemic to Compare Acute and Post-COVID-19 Conditions in Saudi Arabia

**DOI:** 10.3390/jcm12062242

**Published:** 2023-03-14

**Authors:** Aeshah Alsagheir, Samer Amer, Lamya Alzubaidi, Fasial Alenezi, Tareef Alamaa, Abdullah Asiri

**Affiliations:** 1Family and Home Health Care Consultant, Assistant Agency for Hospitals Services, Therapeutic Deputyship, Ministry of Health (MOH), Riyadh 11176, Saudi Arabia; aal-sagheir@moh.gov.sa; 2Department of Public Health and Community Medicine, Zagazig University, Zagazig 44519, Egypt; 3Assistant Agency of Public Health, Ministry of Health (MOH), Riyadh 11176, Saudi Arabia; 4Consultant Diabetologist, Director of Technical Affairs and Ada’a in the Deputy for Hospitals Services, Therapeutic Deputyship, Ministry of Health (MOH), Riyadh 11176, Saudi Arabia; lmalzubaidi@moh.gov.sa; 5Deputy Minister Assistant for Hospitals Services, Therapeutic Deputyship, Ministry of Health (MOH), Riyadh 11176, Saudi Arabia; falenezi@moh.gov.sa; 6Deputy Minister for Therapeutic Services, Therapeutic Deputyship, Ministry of Health (MOH), Riyadh 11176, Saudi Arabia; talama@moh.gov.sa; 7Deputy Minister Assistant for Preventive Services, Consultant of Adult Infectious Diseases, Ministry of Health (MOH), Riyadh 11176, Saudi Arabia; abdullahm.asiri@moh.gov.sa

**Keywords:** post-COVID-19 conditions, acute COVID-19, Saudi Arabia, children, adult, elderly

## Abstract

**Background**: The fight against COVID-19 appears to extend beyond screening and treatment of acute diseases to its medium- and long-term health consequences. Little is known about the epidemiology and the determinants of developing post-COVID-19 conditions (PCCs) among children. The aims of this study were to explore and determine the prevalence of PCCs among three age groups (children and adolescents, adults, and the elderly), and study the predictors of participants’ return to their pre-COVID-19 health status among COVID-19 patients at least four weeks after they got sick, from February to 15 July 2022. **Methods**: This comparison survey study targeted 12,121 COVID-19 patients who fulfilled the selection criteria from the national register system and received a virtual assessment from the Medical Consultation Call Center (937), which was conducted by a well-trained family physician using a validated, well-structured assessment tool. The collected data were coded and analyzed using appropriate tests. **Results:** Out of the 12,121 recovered COVID-19 patients who received the virtual assessment calls, only 5909 (48.8%) agreed and completed the assessment. The majority of participants (4973, or 84.2%) reported no PCCs. The most common PCCs among young people were a cough, dyspnea, fatigue, and loss of appetite or weight loss, while among the elderly they were a cough, dyspnea, fatigue, stomachaches, poor concentration, sleep disturbance, and recurrent fever. Most post-COVID-19 cases require nothing more than reassurance and health education as only 384 (6.5%) required referral to primary health care centers (PHCCs.) The severity of COVID-19 infection, age group, sex, vaccination status, and body mass index were significant predictors for returning to the pre-infection health status and the required referral was significantly related to many factors. **Conclusions:** The comparison of children, adults, and the elderly with regard to the acute and post-COVID-19 conditions in Saudi Arabia in terms of the clinical health assessment and the required management plans showed significant differences.

## 1. Introduction

As of 10 August 2021, after more than nineteen months of this pandemic, the World Health Organization (WHO) has reported more than 203 million confirmed cases of COVID-19 and more than 4.3 million deaths, while in the Kingdom of Saudi Arabia (KSA), there are 533,516 confirmed cases and 8334 deaths [1]. COVID-19 infection can cause multi-organ impairment in people who are at low or high risk of severe acute disease [2]. Collaboration is needed between people who work in public health, politicians, scientists, healthcare services, and the rest of society [3].

The fight against COVID-19 appears to extend beyond screening and treatment of acute disease [4] to include its medium- and long-term health effects. Naming a newly discovered disease is challenging, which includes long-term COVID-19. It has been particularly difficult in the context of the COVID-19 pandemic and the existence of post-COVID-19 conditions (PCCs) [5]. The epidemiological data for PCCs vary by country due to a variety of factors, such as self-reporting accuracy, symptoms examined, length of the follow-up period, differences in healthcare system capability, and reporting system population [6,7,8].

A precise case definition is problematic because, currently, there are no standardized guidelines and there is little consensus on the exact range, prevalence, and duration of symptoms in post-COVID-19 [9]. PCCs are a multisystem disease that includes a wide range of new, returning, or ongoing health problems that people experience after being infected, ranging from mild to incapacitating; even those who were asymptomatic can have PCCs. For varying lengths of time, it has various types and combinations of health problems [10]. Despite the fact that PCCs have predominantly been characterized in adults, concern over PCCs in children has been developing, and among children, they remain poorly described. Clinicians should exercise caution when attributing PCCs following COVID-19 to SARS-CoV-2, even in adults [11,12,13].

Clinical management requires a whole-patient perspective. Despite being a huge burden on healthcare systems worldwide, the volume of published literature describing cases of COVID-19 that later develop PCCs is still small and needs to grow in order to improve our understanding of its epidemiology. To optimize function and quality of life, PCCs should be managed medically through a comprehensive, multidisciplinary management plan based on their patients′ presenting symptoms, underlying medical and mental disorders, and personal and social situations. The treatment goals should be established in cooperation with the appropriate specialists, establishing realistic objectives through cooperative decision making [14,15].

The Kingdom of Saudi Arabia (KSA) was among the first countries to implement early and unprecedented precautionary measures to battle the SARS-CoV-2 pandemic. The KSA is on high alert and prepared to not only take whatever measures are necessary to manage COVID-19 spread but also, on 7 June, the Minister of Hospital Services launched the national Post- COVID-19 Clinical Services across the KSA′s 20 health regions to detect PCCs early and provide effective management services to all inhabitants, reducing the impact on health services and infected individuals [16,17].

The National Institutes of Health (NIH) are researching COVID-19 to improve the recovery (RECOVER) program. So, this research is a top priority of the parent program for this effort [5], which tries to understand, treat, and prevent PCCs using a wide range of research methods, such as electronic health records (EHR), virtual consultations, and real-world data.

Our research supports the WHO and the global initiatives that started over the past year and focused on the three “Rs”: recognition, research, and rehabilitation to advance the understanding of the post-COVID-19 condition in children and young people, which was launched on 17 August 2022. KSA research was limited to adults, a single setting, and exclusively hospitalized patients [18,19,20].

Aim: to improve the overall health status and quality of life of COVID-19 cases in the KSA by providing comprehensive, continuous, evidence-based clinical care to all COVID-19 patients with a rationalized use of resources. Objectives: this study was conducted among COVID-19 patients at least four weeks after the onset of infection, from February to 15 July 2022, in the KSA, to explore and determine the prevalence of PCCs among three age groups (children and adolescents, adults, and the elderly), and study the predictors of participants’ return to their pre-COVID-19 health status.

## 2. Subjects and Methods

### 2.1. Study Design and Setting

A comparative survey through virtual post-COVID-19 clinics at the 937 Call Consultation Center.

### 2.2. Study Population

Target all confirmed diagnosed COVID-19 cases that were classified as positive for SARS-CoV-2 in accordance with the Saudi protocol that was registered in the Health Electronic Surveillance Network (HESN) national registry system, including those who were asymptomatic or mildly to moderately ill with COVID-19, Saudi and non-Saudi, both sexes, and all age groups. If a nucleic acid test on a swab sample from the nares, nasopharynx, or oral cavity was positive, no cases were excluded.

### 2.3. The Assessment (Data Collection) Tool

#### 2.3.1. Assessment (Data Collection), Tool Development, and Structure

Evidence from other studies was incorporated into the survey′s creation and adoption [21,22,23,24,25,26,27,28]. It was initially written in English and then translated into Arabic. A bilingual group of two medical experts and one externally certified medical translator translated the questionnaire′s English version into Arabic. Two English-speaking translators completed the back translation and the original panel was contacted in case there were any problems.

#### 2.3.2. Assessment (Data Collection) Tool Validation

We examined the questionnaire’s content validity, accuracy, and clarity of different items to ensure that the questions were answered consistently. Six different specialists validated the questionnaire. To confirm its comprehension and cultural acceptability, a pilot test was conducted with 400 volunteers from the general community. The participants were asked to rate the questionnaire’s organization, clarity, and length, as well as provide a general opinion. Following that, certain questions were revised in light of their input. To check for reliability and reproducibility, the questionnaire was tested again on the same people one week later. The final data analysis did not include the data collected during the pilot test. We calculated a Cronbach’s alpha of 0.78 for the questionnaire.

#### 2.3.3. The Structure of the Assessment (Data Collection) Tool

(I)Informed consent after giving their permission to take part in the study;(II)The demographic, special habits, comorbidities, and medication;(III)The history and symptoms of acute COVID-19, as well as their classification into critical, severe (managed in hospital), and not severe (managed at home).(IV)Post-COVID conditions are associated with a spectrum of physical, social, and psychological consequences. Check the list for 50 COVID-19-related symptoms [21];(V)The post-COVID-19 clinical assessment through

*The patient self-reported functional status (PCFS) scale*, which is a novel scale that is advised for use during the current epidemic [22]. It evaluates functional aftereffects and tracks immediate improvement four weeks after being discharged from hospital. The PCFS scale encompasses all functional limits over an average of one week.

*To assess reported respiratory disabilities, the Medical Research Council (MRC)* developed the dyspnea scale. Five statements about reported breathlessness are included in the MRC breathlessness scale questionnaire. The grade that applies to them is chosen by the participants. Mild dyspnea was defined as being present in those who chose MRC grades 1 and 2, moderate dyspnea as being present in those who chose grades 3 and 4, and severe dyspnea as being present in those who chose grade 5 breathlessness [23].

*Chronic Fatigue Syndrome Questionnaire (CFS)* [24]. Respondents used an anchored ordinal scale of 0 (no symptom), 1 (trivial), 2 (mild), 3 (moderate), and 4 to score the degree of fatigue and the eight auxiliary criteria over the last six months (severe). Each patient was categorized using the following categories after adding up the eight auxiliary criteria: normal (fatigue = none, trivial, or mild; score of 14), chronic idiopathic fatigue (CIF) (fatigue = moderate or severe; score of 14), CFS-like with insufficient fatigue syndrome (fatigue = none, trivial, or mild; score of 14), and CFS (fatigue = moderate or severe; score of 14) are all different types of fatigue.

*The World Health Organization Five Well-Being Index (WHO-5)* is made up of five positive statements that measure psychological well-being [25]. On a Likert scale, from always (5 points) to never (zero points), participants were asked to score these statements over the previous two weeks (0 points). A total score between 0 and 100 is calculated by multiplying each component of the score by four. A low well-being score of less than 50 shows this.

*Assessment of Generalized Anxiety Disorder-2 (GAD-2).* This is a quick and simple initial screening method. Using a Likert scale of 0—not at all for generalized anxiety disorder, a few days, more than half a day, or almost every day [26].

*Patient Health Questionnaire (PHQ-2 scale).* “Depression screening regarding the prevalence of low mood and anhedonia over the two weeks prior.” The depression Likert scale ranged from 0 to not all, a few, two—more than half the days, three—nearly every day), based on an overall score ranging from 0 to 6 [27].

(VI)The recommended management plans: based on the clinical assessmentNothing; Reassurance; Patient education; Referral to primary care clinics (PHCCs); Referral to post-COVID-19 clinics in hospitals.

### 2.4. The Assessment (Data Collection) Tool Methods

The assessment was conducted by a well-trained family physician through the Medical Consultation Center in the KSA, which offers telephone medical consultations and an e-health call center (937). The Medical Consultation Call Center (937), one of the primary transformation projects of the Saudi Ministry of Health (MOH), is focused primarily on delivering timely medical care and providing each citizen with the necessary medical care [28].

### 2.5. Statistical Analysis

Version 27 of the SPSS software was used to code and analyze the data that was collected. The frequencies (F) and percentages (%) of the COVID-19 symptoms and indicators were calculated at baseline and at least four weeks after infection, using the chi-square (x^2^), Fisher’s exact t-test, and the Mac Nemar test for their analyses. The descriptive statistics were used to summarize the baseline characteristics of the sample after the data in the sub-sample had been cleaned. The mean, median, standard deviation (SD), and interquartile range (IQR) were used to summarize quantitative data. While the f test, analysis of variance (ANOVA), and Kruskal–Wallis test were used for their analyses, the least significant difference (LSD) shows the significance among different groups.

We used simple logistic regression analysis within the framework of generalized linear model approaches to figure out how each important factor was related to the others.

The return to the pre-COVID-19 health level was the post-assessment reference, and the binary result was the baseline factors such as the participants’ age, gender, hospitalizations, comorbidities, and number of symptoms. These were dependent variables. We then used a stepwise approach to design a final logistic regression model to look at the independent relationships between each potential factor and the result of interest.

We first included all the factors that were significant (*p* < 0.05) in the univariable analysis in the stepwise regression technique. Then, in the following steps, we repeatedly assessed each non-significant variable in the final model for potential significance while keeping significant (*p* < 0.05) components in the model. To determine the statistical significance of each factor, we utilized the Likelihood Ratio Tests (LRT).

## 3. Results

### 3.1. As regards the Baseline Demographic of the COVID-19 Cases 

We completed the virtual assessment for (N = 5909). The majority of those who were included were aged between 18 and 65 years (74.5%), and 3227 (54.6%) of them were females. A total of 5276 (89.3%) were Saudis, 3483 (58.9%) were married, 2023 (0.4%) worked in public contact, 3026 (51.2%) had a university degree, 4490 (76.0%) had never smoked, and 2915 (49.3%) had a normal BMI, 649 (11.0%) of the post-COVID-19 participants disclosed a number of comorbidities, and 981 (16.7%) reported that they were on regular medications (Table 1). 

### 3.2. The History of COVID-19 Vaccination, and Self-Reported Acute SARS-CoV-2 Infection Symptoms among Different Age Groups 

There was a statistically significant difference between different age groups and COVID-19 vaccination, as 139 of the elderly (52.5%) received only three doses, while 619 children (49.7%) were unvaccinated, and 2826 adults (64.2%) received only three doses. 

***As regards the severity of infection***, there was a statistically significant difference among the studied groups as regards the post-recovery management received and the severity of infection [Table 2]. 

Fever (4118, 69.7%), cough (2225, 38.2%), sore throat (1674, 28.3%), headache (1431, 24.2%), nasal congestion (1327, 22.5%), and myalgia (1238, 20.9%) were the acute COVID-19 groups. These were the most common acute COVID-19 symptoms in children: fever (986, 79.2%), cough (474, 38.1%), sore throat (399, 32.0%), nasal congestion (298, 23.9%), and myalgia (175, 14.1%). The most common acute COVID-19 symptoms in the elderly were fever (155, 58.5%), cough (112, 43.3%), and sore throat (54, 20.4%). There was a statistically significant difference (*p* < 0.05) in the frequency of all the studied symptoms except night sweats, dizziness, diarrhea, and cough [Table 2].

### 3.3. Self-Reported PCCs among Different Age Groups

The most common PCCs among children and adolescents were cough, dyspnea, fatigue, and loss of appetite or weight loss, while among the elderly they were cough, dyspnea, fatigue, stomachaches, poor concentration, sleep disturbances, and recurrent fever.

There was a statistically significant difference among different age groups and the studied PCCs except for dizziness, joint pain, skin rash, loss of hearing or tetanus, diarrhea, chest pain, and poor concentration, and the self-reported return to the pre-infection health status was significantly higher among the elderly (38.1%) and lower among children and adolescents (5.6%) (Table 3).

### 3.4. The Post-COVID-19 Clinical Assessment 

***As regards respiratory impairments,*** there was a statistically significant difference (*p* < 0.05) among different age groups, and the MRC exertional dyspnea scale was significantly higher among the elderly compared to other groups.

***The WHO-5 questionnaire***: There was a statistically significant difference (*p* < 0.05) among different age groups, and *the WHO-5 questionnaire* had a significantly higher mean ± SD among children and adolescents (75.9 ± 13.1) compared to other groups (Table 4).

***As regards the self-reported CFS***, there was a statistically significant difference (*p* < 0.05) among different age groups, and the CFS was significantly more severe among the lower mean (mean ± SD) and among the children and adolescents median: 1.3 (1.3 ± 1.5) compared to other groups.

***As regards the mental health status****,* there was a statistically significant difference (*p* < 0.05) among different age groups, and the median (mean ± SD) of the total depression and anxiety screening scales was significantly higher among the elderly compared to other groups.

***As regards the self-reported functional status***, after recovering from COVID-19 disease, there was a statistically significant difference (*p* < 0.05) among different age groups, and the PCFS was significantly more severe among the elderly compared to other groups (Table 4).

### 3.5. Post-COVID-19 Clinical Management

There was a statistically significant difference (*p* < 0.05) among different age groups and the required management plans as, for example, the referral to PHCCs was significantly higher among adults (14.3%) compared to other groups, while among children and adolescents the reassurance was 40% compared to other groups (Figure 1).

### 3.6. Post-COVID-19 Infection, Several Factors Predict a Recovery to Baseline Health Status and Require Referral to PHCCs and Post-COVID-19 Clinics in Hospitals 

The findings of the univariate and multivariate analyses for determinants of the return to baseline health status are summarized in Table 5. Young age, female sex, smoking, comorbidities, body mass index, and severity of acute COVID-19 symptoms were all found to be negative predictors of return to baseline health status on univariate analysis (*p* < 0.05 for all comparisons). We then used multivariable logistic regression analysis to identify the independent risk factors associated with participants’ return to their pre-COVID-19 disease baseline health status.

## 4. Discussion

This study is a comparative national assessment survey of children, adults, and the elderly in the fourth wave of the COVID-19 pandemic in the KSA. It is a comparison to address the issue of concern for acute and PCCs to cover all infected patients, symptomatic and asymptomatic, using a self-reported and validated clinical assessment tool, so this study provides important and generalizable evidence.

### 4.1. The Self-Reported Acute and PCCs among Different Age Groups

There was a statistically significant difference among different age groups and the studied PCCs except for dizziness, joint pain, skin rash, loss of hearing or tetanus, diarrhea, chest pain, and poor concentration, and the self-reported return to the pre-infection health status was significantly higher among the elderly (38.1%) and lower among children and adolescents (5.6%). There was a statistically significant difference among the studied groups as regards the post-recovery management received and the severity of infection.

The most common self-reported PCCs among children (less than 18 years old) after more than 4 weeks from the acute infection were a cough, dyspnea, fatigue, and loss of appetite or weight loss. According to a large study from 8 countries conducted between March 2020 and January 2021, the most common symptoms after 90 days after the acute infection were fever (1241, 65.9%); cough (917, 48.7%); and rhinorrhea or congestion (893, 47.4%). In other studies, fatigue, headaches, and not being able to smell have been found to be common PCC symptoms in children. A large United Kingdom (UK) study found the same thing [29,30,31]. This could be related to a symptom that, throughout the study period, had a significant impact on deciding access to testing. Thus, local testing standards may have an impact on PCC symptoms.

The prevalence of self-reported PCCs was 7.9% among youth less than 18 years old after 28 days or more of follow-up, 5% of the 175 children in the United Kingdom, and only 4% of the children in the UK. Moreover, in subsequent research on Russian children after more than five months, lower PCC rates were observed, with 25% reporting that PCCs had the greatest early report. In 8 countries from March to December 2021, only 44 (or 9.8%) of hospitalized children had PCCs. Sixty percent of asymptomatic children had a three-month follow-up [32]. Australia’s 3- to 6-month follow-up was 12%. The various figures offered by the studies cited above most likely reflect differences in response rates, PCC criteria, and methods for detecting PCCs. The reduced PCC prevalence could be attributed to the shorter follow-up duration, the restricted vaccination policy in the KSA, the virtual assessment, and the children’s self-reporting of PCCs after parental consent. Their younger, less-verbal children are also less likely to report specific symptoms compared with verbal teenagers. A consequent variable risk of bias [31,32,33,34,35,36].

There was a statistically significant difference between age groups and the studied acute COVID-19 symptoms (all PCCs except dizziness, joint pain, skin rash, loss of hearing or tetanus, diarrhea, chest pain, and poor concentration), the required management referral, and the self-reported return to pre-infection health status, with the elderly (38.1%) being significantly higher and children and adolescents (5.6%) being significantly lower. This can be explained by the fact that the risk of severe illness from COVID-19 increases with age. The elderly were at high risk of developing COVID-19 with rapidly deteriorating medical symptoms. In fact, cytokine storms caused by viruses are more likely to happen in older people because their immune systems are weaker and they have other health problems. This can affect multiple systems and cause life-threatening respiratory failure [37].

### 4.2. The Post-COVID-19 Clinical Assessment 

Despite the prevalence of self-reported PCCs being reported by 936 people (15.8%), after the professional clinical assessment reported abnormal CFS being reported by 11.4% and functional limits being reported by 5.3%, 1931 people (33.5%) received a well-being score below 50, which denotes poor wellbeing. Furthermore, 10 people (0.2%) had severe dyspnea, whereas 5822 (98.5%) had mild dyspnea, 76 (1.3%) had moderate dyspnea, 52 (0.8%) had an anxiety disorder, and 63 (1.1%) had a depression disorder, in agreement with previous research that revealed that the findings of professional evaluation and self-rating differed. Those differences may be associated with the personality and demography of the patients, which is consistent with the previous studies. Because of this, it is crucial to accurately assess patients’ health status, which can prevent the waste of scarce health resources and enable patients to get care using less strenuous methods. On the contrary, it might lighten the workload for medical personnel. While professional evaluation is still the gold standard for health disorders, it is a comprehensive evaluation of the patient’s symptoms by clinically trained professionals with experience [37,38,39,40].

As regards CFS, because viral infections have been connected to CFS [41], we used the eight-item CFS questionnaire to investigate the nature of exhaustion in the PCCs and reported abnormal CFSs being reported among 11.4%, which is significantly lower than the reported frequency among other different studies to be (SA, December 2021) 38.4% for at least 4 weeks [20,42]. On the other hand, 25 of 29 CFS symptoms were reported by at least one COVID-19 study in a recent meta-analysis that addressed CFS symptoms in COVID-19 disease and included 21 studies [43]. Additionally, following the recovery from SARS, similar findings to ours have been published. According to Lam et al.’s report, 27.1% of COVID-19 patients who recovered met the criteria for CFS [41].

Shortness of breath was reported by 23.9% of participants, and joint problems were recorded by 30.5% of individuals, among other significant findings. More than 20% of patients have also mentioned other symptoms, such as headaches, dizziness, loss of taste and smell, sleeplessness, appetite loss, and difficulty concentrating. The persistence of many symptoms in our patients is comparable to what Davis et al. found in an international cohort study in which they tracked 66 symptoms over the course of 7 months following the onset of the COVID-19 disease [44]. A recent meta-analysis of 15 trials with 47,910 patients and 50 long-term symptoms of COVID-19 disease backed up these findings. The most common long-term symptoms were fatigue (58%), headaches (44%), attention problems (27%), hair loss (25%), and shortness of breath (24%).

We did univariate and multivariate analyses to find out what factors were linked to getting better after SARS-CoV-2 infection. We found a few independent predictors that were linked to getting better. Female sex was associated with a delayed return to pre-illness status, with an OR of 0.72 (95% CI 0.63–0.82, *p* = 0.001). Numerous investigations [45,46,47,48] have universally documented this observation. The constant observation of a higher likelihood of developing post-COVID-19 syndrome in females is not totally understood; however, it may be partially due to the known sex dimorphism seen with some disorders, such as autoimmune diseases [29]. 

Unexpectedly, there was a consistent rise in the odds of returning to the pre-illness baseline condition for every 10 years older when we used the age group 25–34 as a reference. The highest OR was seen in individuals over 54 years old (OR 2.60, 95% CI 1.59–4.25, *p* = 0.001). All other research, as well as our earlier conclusion from a smaller cohort of hospitalized COVID-19 patients, are in conflict with this finding. This could be the result of selection bias, as only a specific group of elderly patients in good health were eligible to participate in this online survey. It might also be the result of chance or other unknown causes. However, more research needs to be carried out on this in the future. We also demonstrated that failure to return to the pre-illness baseline state was associated with hospital admission (OR 0.35, CI 0.21–0.59, *p* = 0.001), the number of symptoms (OR 0.91, CI 0.89–0.92, *p* = 0.001), and several comorbidities such as arrhythmias and asthma. These results concur once more with those of other studies [15,45,46,47,48].

### 4.3. The Post-COVID-19 Clinical Management

The overall referral rate of PCCs to PHCCs was 6.5%, which is higher than the average telephone consultation referral rate in Saudi Arabia, which represents about 2% of primary healthcare center visits. This may be attributed to the fact that these services are proactive primitive services, which the majority of participants believed [49]. In agreement with another study in 8 countries between March 2020 and January 2021, it was discovered in this study that PCCs were reported at 90 days by 9.8% of SARS-CoV-2-infected hospitalized children and 4.6% of SARS-CoV-2-infected non-hospitalized children. Comparable hospitalized children without SARS-CoV-2 infections reported comparable symptoms at 5.0% and 2.7%, respectively. PCCs symptoms: the length of the hospital stay, the number of symptoms present, and advanced age were risk variables linked to PCCs [29].

Fortunately, some countries have started implementing clinical guidelines to help clinicians, including SA. In May 2022, the Saudi MOH published a Saudi guideline for post-COVID-19 clinical care. It is essential to implement guidelines for long-term COVID management [50]. As this disease continues to spread and an increasing number of people experience chronic persistence of COVID-19 symptoms, the healthcare system will face economic, organizational, and structural difficulties. A well-defined, lengthy COVID management protocol will greatly benefit the overburdened healthcare system in the near future [51].

### 4.4. Strengths and Limitations 

This is a top priority of the parent program for the NIH Researching COVID to Improve Recovery Program, which attempts to understand, treat, and prevent PCCs. This research gives information about possible subtypes and current practice patterns related to PCCs. It also shows that there are often differences in how people with PCCs are diagnosed. Naming diseases is always a challenge, and there are a lot of efforts to standardize, clear up, and keep track of the names and definitions of diseases.

Our study does, however, have some drawbacks. First, all observational research flaws, including bias and confounding, exist in our study. Second, because of recall bias and availability bias, especially for the age group younger than 18 years, and because this was not a prospective study, which lowers the accuracy of the data collected. Third, individuals from an earlier wave of the COVID-19 condition were included in our study. The reporting of pertinent facts may have been impacted by the lack of knowledge regarding the long-term effects of COVID-19 during this time. Fourth, because we did not do an evaluation of the pre-COVID-19 baseline health, it is impossible to distinguish between pre-existing issues and those connected to the COVID-19 condition. Finally, the study lacked external validation. Despite these drawbacks, evidence from other foreign studies lends support to our conclusions.

## 5. Conclusions

The comparison of children, adults, and the elderly as regards the acute, PCC symptoms, clinical health assessment, and the required management plans showed significant differences. The severity of SARS-CoV-2 infection, age group, sex, vaccination status, and body mass index were significant predictors for returning to the health level before the infection and the need for a referral were linked by many factors.

## 6. Recommendation

(1) Informing COVID-19 outpatient patients of potential long-term COVID-19 consequences is what we advised. (2) Doctors should be aware of the various causes of symptoms such as fatigue, cognitive and neurologic symptoms, and dyspnea, and look for a differential diagnosis to avoid misinterpretation in order to better manage and assist those who are more at risk for persistent symptoms. (3) The establishment of an outpatient clinical setting for post-COVID-19 in order to evaluate, monitor, support, and manage recovered cases. (4) Developing a post-COVID-19 management protocol and guideline that is well-structured and based on the national evidence-based findings. (5) Additional research is needed to close any gaps or omissions in the clinics or the procedure. Long-term studies are also required to investigate the etiology of the disease and demonstrate the effectiveness of the treatments.

## Figures and Tables

**Figure 1 jcm-12-02242-f001:**
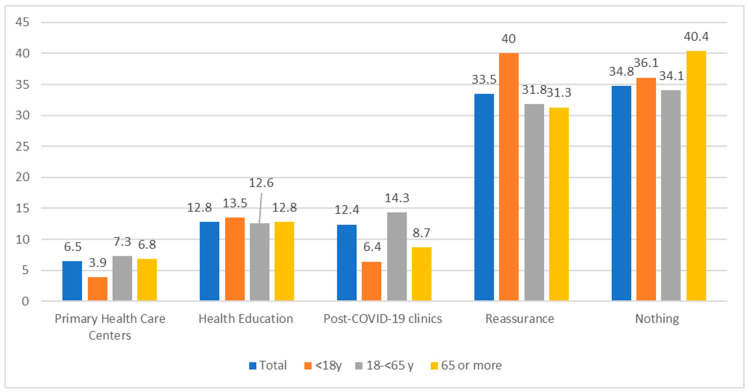
The required management plan among different ages of post-COVID-19 cases. X^2^ = 91.9 (0.00 *), ** p* < 0.05 there was a statistically significant difference.

**Table 1 jcm-12-02242-t001:** Demographic of general characteristics of the COVID-19 participants, and its relationship among different age groups.

	<18 Y T = 1245 F (%)	18–65 y T = 4399 F (%)	≥65 y T = 265 F (%)	(*p*)
Age (y)	11	37	70	
Median (mean ± SD)	(1–17)	(18–64)	(65–98)	(0.00 *)
Nationality				
Saudi	1150(92.4)	3880(88.2)	246(92.8)	
Non-Saudi	95(7.6)	519(11.8)	19(7.2)	(<0.001 *)
Marital status				
Child	501(40.2)	0(0.0)	0(0.0)	
Single	731(58.7)	1058(24.1)	1(0.4)	
Married	13(1.0)	3224(73.3)	246(92.8)	(<0.001 *)
Divorced/widow	0(0.0)	117(2.7)	18(6.8)	
Occupation				(<0.001 *)
Child	355(28.5)	0(0.0)	0(0.0)	
Student	874(70.2)	298(6.7)	0(0.0)	
Housewife/not working/retired	12(1.0)	1101(25.0)	235(88.7)	
Work in contact with public	1(0.1)	2011(45.7)	12(4.5)	
Don’t work in contact with public	3(0.2)	739(16.8)	17(6.4)	
Level of education				(<0.001 *)
Child	303(24.3)	0(0.0)	0(0.0)	
Illiterate	15(1.2)	67(1.5)	25(9.4)	
Primary/read and write	407(32.7)	234(5.3)	106(40.0)	
Preparatory	224(18.0)	232(5.3)	16(6.0)	
Secondary	286(23.0)	903(20.5)	18(6.8)	
University or higher	11(0.9)	2963(67.4)	53(20.0)	
Sex				
Male	601(49.3)	1921(43.7)	149(56.2)	(<0.001 *)
Female	644(51.7)	2477(56.3)	116(43.8)	
Smoking				
Not-smokers	1147(6.80	3125(71.0)	218(82.3)	
Passive smokers	85(6.8)	220(5.0)	7(2.6)	
Smokers	13(1.0)	927(21.1)	16(6.0)	(<0.001 *)
Ex-smoker	0(0.0)	127(2.9)	24(9.1)	
BMI (kg/m^2^)				(<0.001 *)
Underweight	96(7.7)	128(2.9)	7(2.6)	
Normal	873(70.1)	1933(43.9)	109(41.1)	
Overweight	95(7.6)	815(18.5)	25(9.4)	
Obese	10(0.8)	38(0.9)	1(0.4)	
Don’t know/not sure	171(13.7)	1485(33.9)	123(46.4)	
Co-morbidity	35(2.8)	470(10.7)	144(54.3)	(0.00 *)
On regular medications	47(3.8)	765(17.4)	169(63.8)	(0.00 *)

* *p* < 0.05 there was a statistically significant difference. There was a statistically significant difference (*p* < 0.05) between the studied different age groups and all the studied demographic data.

**Table 2 jcm-12-02242-t002:** The history of COVID-19 vaccination, and acute SARS-Cov2 infection among different age groups.

	<18 y T = 1245 F (%)	18–65y T = 4399 F (%)	≥65 y T = 265 F (%)	*(p)*
COVID-19 vaccination				(<0.001 *)
Unvaccinated	619(49.7)	80(1.8)	17(6.4)	
Received only one dose	35(2.8)	65(1.5)	12(4.5)	
Received only two doses	525(42.2)	2826(64.2)	91(34.3)	
Received only three doses	66(5.3)	1417(32.2)	139(52.5)	
Received four doses	1(0.1)	11(0.3)	6(2.3)	
The acute COVID-19 symptoms				(<0.001 *)
Asymptomatic	127(10.2)	388(8.8)	41(15.5)	
Symptomatic	1118(89.8)	4011(91.2)	224(84.5)	
Acute COVID-19 symptoms				
Fever	986(79.2)	2977(67.7)	155(58.5)	
Nasal congestion	399(32.0)	896(20.4)	32(12.1)	(<0.001 *)
Fatigue	79(6.3)	667(15.2)	20(7.5)	(<0.001 *)
Sore throat	298(23.9)	1322(30.1)	54(20.4)	(<0.001 *)
Nausea and vomiting	62(4.9)	115(2.6)	5(1.9)	(<0.001 *)
Dyspnea	93(7.5)	561(12.8)	50(18.9)	(<0.001 *)
Cough	474(38.1)	1668(37.9)	115(43.4)	(<0.001 *)
Myalgia	95(7.6)	1094(24.9)	49(18.5)	(0.20)
Dizziness	23(1.8)	143(3.3)	8(3.0)	(<0.001 *)
Diarrhea	37(2.9)	108(2.5)	4(1.5)	(0.33)
Loss of appetite	13(1.04)	27(0.6)	4(1.5)	(0.09)
Loss of smell	20(1.6)	285(6.5)	8(3.0)	(<0.001 *)
Loss of taste	32(2.3)	371(8.4)	10(3.8)	(<0.001 *)
Headache	175(14.1)	1226(27.9)	30(11.3)	(<0.001 *)
Stomachache	18(1.4)	40(0.9)	7(2.6)	(0.01 *)
Chest pain	0(0.0)	103(2.3)	6(2.3)	(<0.001 *)
Night sweats	2 (0.2)	16(0.4)	0(0.0)	(0.53)
Conjunctivitis	2(0.2)	14(0.3)	0(0.0)	(<0.001 *)
Blurring of vision	0(0.0)	0(0.0)	5(1.9)	(0.02 *)
Poor concentration	0(0.0)	5(0.11)	2(0.8)	(0.76)
Others	10(0.8)	33(0.8)	1(0.4)	
Complications				(0.89)
No	1244(99.9)	4394(99.9)	260(98.1)	
Pulmonary complications	1(0.1)	0(0.0)	3(1.2)	
Cardiovascular complications	0(0.0)	2(0.0)	1(0.4)	
Neurological complications	0(0.0)	3(0.0)	0(0.0)	
Psychiatric complications	0(0.0)	0(0.0)	1(0.4)	
Severity of infection				(0.00 *)
Critical	6(0.5)	11(0.3)	6(2.3)	
Severe (managed at hospital)	15(1.2)	32(0.7)	25(9.4)	
Not severe (managed at home)	1224(98.3)	4356(99.0)	234(88.3)	
Post-recovery management				(0.00 *)
Nothing	1227(98.6)	4370(99.3)	249(94.0)	
Re-hospitalized	4(0.3)	13(0.3)	0(0.0)	
Follow up plan	11(0.9)	15(0.3)	15(5.7)	
Referred to PHCCs	3(0.2)	1(0.0)	1(0.4)	

* *p* < 0.05 there was a statistically significant difference; PHCCs (primary health care centers).

**Table 3 jcm-12-02242-t003:** Self-reported post-COVID-19 symptoms among different age groups.

	<18 y T = 1245 F (%)	18–65 T = 4399 F (%)	≥65 y T = 265 F (%)	*(p)*
Post-COVID-19 symptoms				
Nothing	1146(92.0)	3612(82.1)	215(81.1)	
Cough	38(3.1)	191(4.3)	17(6.4)	(<0.001 *)
Dizziness	3(0.24)	23(0.5)	0(0.0)	(0.03 *)
Dyspnea	23(1.8)	176(4.0)	10(3.8)	(0.42)
Loss of appetite/loss of weight	15(1.2)	25(0.6)	3(1.1)	(0.001 *)
Nausea/vomiting	1(0.08)	12(0.3)	3(1.1)	(0.04 *)
Pain	0(0.0)	138(3.1)	9(3.4)	(0.01 *)
Chest pain	3(0.24)	29(0.7)	2(0.8)	(<0.001 *)
Stomachache	5(0.4)	91(2.1)	5(1.9)	(0.21)
Joint pain	3(0.2)	10(0.22)	1(0.4)	(<0.001 *)
Fatigue	19(1.5)	173(3.9)	9(3.4)	(0.88)
Sleep disturbances	10(0.8)	103(2.3)	3(1.1)	(<0.001 *)
Diarrhea	1(0.00)	10(0.22)	1(0.4)	(0.002 *)
Loss of smell	4(0.3)	75(1.7)	1(0.4)	(0.484)
Loss of taste	0(0.0)	0(0.0)	1(0.4)	(<0.001 *)
Loss of hearing/tetanus	2(0.2)	6(1.4)	1(0.4)	----
Headache	5(0.4)	76(1.7)	0(0.0)	(0.61)
Recurrent fever	5(0.4)	4(0.9)	3(1.1)	(<0.001 *)
Memory impairment	5(0.4)	129(2.9)	00(0.0)	(0.364)
Poor concentration	8(0.7)	38(0.9)	4(1.6)	(0.001 *)
Psychiatric impairment	8(0.7)	93(2.1)	2(0.8)	(0.02 *)
Hair loss	6(0.4)	60(1.4)	1(0.4)	(0.51)
Skin rash	2(0.2)	5(0.1)	0(0.0)	(0.02 *)
Menstrual disturbances	3(0.2)	44(1.0)	0(0.0)	(<0.001 *)
Others	7(0.6)	31(0.7)	19(7.2)	
Return to the pre-infection health status				
Resolved	1175(94.4)	3899(88.6)	227(85.7)	
Unresolved	70(5.6)	500(11.4)	38(14.3)	(0.00 *)
Duration to return to the pre-infection health status (d)				
Median (IQR)	5(0–30)	7(0–71)	7.9(0–74)	(0.00 *)

* *p* < 0.05 there was a statistically significant difference. d (days).

**Table 4 jcm-12-02242-t004:** Post-COVID-19 symptoms among different age groups.

	<18 y T = 1245 F (%)	18–65 y T = 4399 F (%)	≥65 y T = 265 F (%)	*(p)*
Medical research council (MRC) dyspnea scale				
Mild dyspnea	1231(98.9)	4335(98.5)	256(96.7)	(<0.001 *)
Moderate dyspnea	13(1.0)	59(1.4)	4(1.5)	
Severe dyspnea	0(0.0)	5(0.1)	5(1.9)	
None	1(0.1)	0(0.0)	0(0.0)	
WHO-5 well-being score				
Mean (SD)	75.9(13.1) a	65.9(13.9) b	60.9(24.3) c	(0.00 *)
Chronic fatigability				
No/unapplicable	1155(92.8)	3669(83.4)	219(82.6)	
Short-term memory impairment	41(3.3)	306(6.9)	19(7.2)	
Sore throat	25(2.0)	82(0.9)	10(3.8)	
Lymph node	16(1.3)	24(0.5)	5(1.9)	
Muscle pain	36(2.9)	225(5.1)	54(20.3)	
Difficult sleep	36(2.9)	200(4.5)	52(19.6)	
Extreme fatigability	42(3.4)	271(5.8)	75(28.3)	
Joint pain	35(2.9)	231(5.3)	63(23.8)	
Chronic fatigability scale				
Median (IQR)	1a(0–15)	1b(0–21)	2b(0–22)	(0.00 *)
Depression (PHQ-2)				
Median (IQR)	0(0–6)	0(0–6)	0(0–6)	(0.00 *)
Anxiety (GAD-2)				
Median (IQR)	0(0–6)	0(0–6)	0(0–6)	(0.00 *)
Patient self-report functional status (PCFS) scale	1216(97.7)	4132(93.3)	248(93.6)	(<0.0001 *)
No functional status limits	0(0.0)	0(0.0)	0(0.0)	
Little functional status limits	1(0.1)	13(0.3)	5(1.9)	
Mild restriction	10(0.8)	127(2.9)	4(1.5)	
Moderate restriction	2(0.2)	11(0.3)	3(1.1)	
Severe restriction	16(1.3)	116(2.6)	5(1.9)	

* *p* < 0.05 there was a statistically significant difference; a, b, c, the different symbols showed a statistically significant difference.

**Table 5 jcm-12-02242-t005:** Factors significantly associated with the participants’ reports of delayed return to pre-COVID-19 infection baseline health status.

Variables	Univariable Analyses		Multiple Logistic Regression Model (Adjusted ORs)	
Demographics	OR (95% CI)	*p*	OR (95% CI)	*p*
Age (groups)				
Adult (18 y to less than 65 y)	Reference	-	-	-
Less than 18 y	0.50 (0.34–074)	0.025 *	0.74 (0.62–0.88)	0.020 *
65 y or more	12.68 (1.71–4.20)	<0.001	12.63 (1.64–4.23)	<0.001
Female gender	0.55 (0.48–0.61)	<0.001	0.72 (0.62–0.82)	<0.001
Severity of SARS-CoV-2 infection	4.83 (3.72–6.97)	0.023 *	1.93 (1.78–5.10)	0.372
Smoking status				
Never smoked	Reference	-	-	-
Ex-smoker	1.22 (1.00–1.49)	0.050	-	-
Current smoker	1.23 (1.05–1.45)	0.012 *	-	-
Unknown	1.01 (0.60–1.69)	0.975	-	-
Vaccination status				
Unvaccinated				
Received only one dose	3.7 (1.34–6.74)	<0.001 *	1.75 (1.48–2.16)	0.001 *
Received only two doses	0.84 (0.77–0.91)	<0.001 *	0.60 (0.46–0.77)	<0.001
Received only three doses	0.27 (0.08–0.94)	0.040 *	0.47 (0.12–1.81)	0.274
Received four doses	References	--	--	-
BMI (kg/m^2^)	1.97 (0.99–3.11)	<0.001 *	1.91 (1.00–2.2)	<0.001
Comorbidities	3.7(1.9–11.7)	<0.001 *	2.1(1.0–3.6)	<0.001

* *p* < 0.05 there was a statically significant difference.

## Data Availability

The data are available when requested from the corresponding author, dr_samar11@yahoo.com.

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
