# Peer review of "A National Survey of Children, Adults, and the Elderly in the Fourth Wave of the COVID-19 Pandemic to Compare Acute and Post-COVID-19 Conditions in Saudi Arabia"

_jcm, 2023, doi:10.3390/jcm12062242_

Round 1

Reviewer 1 Report

This is an interesting and well conducted study that is worthy of publication.  It does require some revision.  Specific suggestions below.

There is text missing from line 37 of the abstract making the sentence ". require nothing more than reassurance and health education only 384 37 (6.5 %) required referral to PHCCs" uninterpretable.

Line 93 remove so from this sentence "; So this 93 study was conducted among COVID-19 patients at..."

this part of the same sentence doesn't make sense and needs to be revised ": to determine the prevalence of PCCs, to 95 compare in an age-stratified analysis of three age groups (children and adolescent , adults, 96 and old age), and to participants’ return to pre-COVID-19 health status and associated 97 risk factor. 98"

Please review this sentence. It doesn't quite make sense. "There was a statistically significant difference between different age groups and 214 COVID-19 vaccination, as 139(52.5%) of the elderly received three doses, while 619(49.7%) 215 of children were unvaccinated, and 2826(64.2%) of adults received only three doses."  

Line 317 - full stop missing

Line 318-9 - "25% of them reported that 318 PCCs had the greatest early report." This sentence doesnt make sense

The final paragraph "recommendation" is poorly written and should be revised in my opinion.

Author Response

There is text missing from line 37 of the abstract making the sentence ". require nothing more than reassurance and health education only 384 37 (6.5 %) required referral to PHCCs" uninterpretable.

Line 93 removes so from this sentence "; So this 93 study was conducted among COVID-19 patients at..."done

this part of the same sentence doesn't make sense and needs to be revised ": to determine the prevalence of PCCs, to 95 compare in an age-stratified analysis of three age groups (children and adolescent , adults, 96 and old age), and to participants’ return to pre-COVID-19 health status and associated 97 risk factor. 98"----------done

Please review this sentence. It doesn't quite make sense. "There was a statistically significant difference between different age groups and 214 COVID-19 vaccination, as 139(52.5%) of the elderly received three doses, while 619(49.7%) 215 of children were unvaccinated, and 2826(64.2%) of adults received only three doses." ----------done 

Line 317 - full stop missing----corrected 

Line 318-9 - "25% of them reported that 318 PCCs had the greatest early report." This sentence doesn't make sense------------corrected 

The final paragraph "recommendation" is poorly written and should be revised in my opinion.--removed 

Reviewer 2 Report

The manuscript by Aeshah Alsagheir et al. entitled “A National Survey of Children, Adults, and the Elderly in the Fourth Wave of the COVID-19 Pandemic to Compare Acute and Post-COVID-19 Conditions” aimed to determine the prevalence of PCCs, to compare in an age-stratified analysis of three age groups (children and adolescent , adults, and old age), and to participants’ return to pre-COVID-19 health status and associated risk factor.

The abstract summarizes the general significance of the manuscript and the article leads some evidence to such point, but there  some minor issues need to be addressed to improve the significance of the manuscript:

--Firstly, the exclusion criteria are not clear.

--Moreover, the bibliography should be expanded; accordingly, these articles should be cited:

-Pfaff ER, Madlock-Brown C, Baratta JM, et al. Coding long COVID: characterizing a new disease through an ICD-10 lens. BMC Med. 2023;21(1):58. Published 2023 Feb 16. doi:10.1186/s12916-023-02737-6

-Visco V, Vitale C, Rispoli A, Izzo C, Virtuoso N, Ferruzzi GJ, Santopietro M, Melfi A, Rusciano MR, Maglio A, Di Pietro P, Carrizzo A, Galasso G, Vatrella A, Vecchione C, Ciccarelli M. Post-COVID-19 Syndrome: Involvement and Interactions between Respiratory, Cardiovascular and Nervous Systems. J Clin Med. 2022 Jan 20;11(3):524. doi: 10.3390/jcm11030524. PMID: 35159974; PMCID: PMC8836767.

Author Response

There is text missing from line 37 of the abstract making the sentence ". require nothing more than reassurance and health education only 384 37 (6.5 %) required referral to PHCCs" uninterpretable.

Line 93 remove so from this sentence "; So this 93 study was conducted among COVID-19 patients at..."

this part of the same sentence doesn't make sense and needs to be revised ": to determine the prevalence of PCCs, to 95 compare in an age-stratified analysis of three age groups (children and adolescent, adults, 96 and old age), and to participants’ return to pre-COVID-19 health status and associated 97 risk factor. 98"

Please review this sentence. It doesn't quite make sense. "There was a statistically significant difference between different age groups and COVID-19 vaccination, as 139(52.5%) of the elderly received three doses, while 619(49.7%)  children were unvaccinated, and 2826(64.2%) adults received only three doses."---------  

Line 317 - full stop missing-----done 

Line 318-9 - "25% of them reported that 318 PCCs had the greatest early report." This sentence doesn't make sense----------done 

The final paragraph "recommendation" is poorly written and should be revised in my opinion-----------removed 

The manuscript by Aeshah Alsagheir et al. entitled “A National Survey of Children, Adults, and the Elderly in the Fourth Wave of the COVID-19 Pandemic to Compare Acute and Post-COVID-19 Conditions” aimed to determine the prevalence of PCCs, to compare in an age-stratified analysis of three age groups (children and adolescent, adults, and old age), and to participants’ return to pre-COVID-19 health status and associated risk factor.--------corrected 

The abstract summarizes the general significance of the manuscript and the article leads some evidence to such point, but there are some minor issues need to be addressed to improve the significance of the manuscript:--------corrected 

--First, the exclusion criteria are not clear.----------no cases were excluded

--Moreover, the bibliography should be expanded; accordingly, these articles should be cited:

-Pfaff ER, Madlock-Brown C, Baratta JM, et al. Coding long COVID: characterizing a new disease through an ICD-10 lens. BMC Med. 2023;21(1):58. Published 2023 Feb 16. doi:10.1186/s12916-023-02737-6---------added 

-Visco V, Vitale C, Rispoli A, Izzo C, Virtuoso N, Ferruzzi GJ, Santopietro M, Melfi A, Rusciano MR, Maglio A, Di Pietro P, Carrizzo A, Galasso G, Vatrella A, Vecchione C, Ciccarelli M. Post-COVID-19 Syndrome: Involvement and Interactions between Respiratory, Cardiovascular and Nervous Systems. J Clin Med. 2022 Jan 20;11(3):524. doi: 10.3390/jcm11030524. PMID: 35159974; PMCID: PMC8836767.--------------added
